# Single-Shot Intrinsic Calibration for Autonomous Driving Applications

**DOI:** 10.3390/s22052067

**Published:** 2022-03-07

**Authors:** Abraham Monrroy Cano, Jacob Lambert, Masato Edahiro, Shinpei Kato

**Affiliations:** 1Graduate School of Information Science, Nagoya University, Nagoya 464-8603, Japan; jacob.lambert@g.sp.m.is.nagoya-u.ac.jp (J.L.); eda@ertl.jp (M.E.); 2Graduate School of Information Science and Technology, The University of Tokyo, Tokyo 113-8656, Japan; shinpei@pf.is.s.u-tokyo.ac.jp

**Keywords:** cameras, intrinsic, calibration, robotics, LiDAR, sensors, autonomous driving

## Abstract

In this paper, we present a first-of-its-kind method to determine clear and repeatable guidelines for single-shot camera intrinsic calibration using multiple checkerboards. With the help of a simulator, we found the position and rotation intervals that allow optimal corner detector performance. With these intervals defined, we generated thousands of multiple checkerboard poses and evaluated them using ground truth values, in order to obtain configurations that lead to accurate camera intrinsic parameters. We used these results to define guidelines to create multiple checkerboard setups. We tested and verified the robustness of the guidelines in the simulator, and additionally in the real world with cameras with different focal lengths and distortion profiles, which help generalize our findings. Finally, we used a 3D LiDAR (Light Detection and Ranging) to project and confirm the quality of the intrinsic parameters projection. We found it possible to obtain accurate intrinsic parameters for 3D applications, with at least seven checkerboard setups in a single image that follow our positioning guidelines.

## 1. Introduction

Navigation robots, such as autonomous vehicles, require a highly accurate representation of their surroundings to navigate and reach their target safely. Sensors such as cameras, radars, and LiDARs (Light Detection and Ranging) are commonly used to provide rich perception information. Each of these sensors can complement each other to supply reliable and accurate data. For example, cameras produce a dense representation of the world, including color, texture, and shape. However, cameras cannot provide reliable depth information at longer distances. On the other hand, LiDARs capture dense and highly accurate range information at short, middle, and often at long range regardless of the lighting conditions.

The simultaneous integration of data from multiple sensors is known as fusion, and it is used to overcome weaknesses in each individual sensor. State-of-the-art perception algorithms utilize fused data inside deep neural networks to improve detection accuracy. For example, some of these networks require the image, the point cloud data, and accurate camera intrinsic and camera-lidar extrinsic parameters to enable training and inference [1,2,3,4]. Another common application that requires precise calibration is camera-based localization, also known as visual SLAM (Simultaneous Localization and Mapping) [5,6]. On the other hand, applications that do not require fusion and only operate on images might not be significantly affected by small errors in the intrinsic camera parameters. Recent advances in deep learning [7,8] apply data augmentation techniques to increase resilience to image distortions. Regardless of whether camera data is used independently or as part of a fusion methodology, any application involving 3D geometry will require accurate and careful sensor calibration. Fundamental to this is how a camera is modeled in terms of its intrinsic parameters.

Cameras have become ubiquitous thanks to their low cost, high quality, and ability to represent the world with dense and feature-rich images. The images created by these devices resemble our own vision, depicting objects located at different distances with different apparent dimensions. The mathematical model commonly used to project the three-dimensional world is the pin-hole camera model. In addition, the plumb-bob model, also known as the Brown–Conrady model, represents the distortion caused by the lens attached to the camera [9]. Model parameters can be estimated using a method known as camera calibration (also referred to as geometric camera calibration or camera re-sectioning). This method requires capturing images while moving either the camera itself or a calibration target, with identifiable features of known dimensions, aiming to cover the entire camera’s field of view. The targets used in this calibration process depend on the selected algorithm. Methods such as the one presented by Zhang [10], use one-dimensional targets in the form of a stick with beads attached to it separated by a known distance. Two-dimensional target arrays have a low cost, and the methods developed for this modality provide sub-pixel accuracy. Finally, three-dimensional targets, often used for photogrammetry applications [11], offer higher accuracy, but are not easily obtainable. For the previous reasons, computer vision and robotics applications traditionally employ 2D planar grid targets in the form of large flat boards with identifiable patterns such as checkerboards, arranged circles, or fiducial markers, among others. Correspondences between feature points on the planar target among all of the frames are determined in order to calculate the intrinsic parameters. This process can be tedious, especially on robots or vehicles featuring large arrays of cameras.

There are instructions on performing the data capture procedure, such as covering the entire frame, making sure the focus is correct, taking multiple images while keeping the focus and focal length fixed, or locating the target at the same distance as the measure of the planned object [12,13,14,15]; however, there are no clear instructions on setting the checkerboard pose to facilitate the process while also obtaining accurate parameters. For this reason, after finishing the data acquisition and estimating the parameters, their validity is not apparent until they are applied to project 3D points. The metric commonly used to evaluate the accuracy of the parameters is the re-projection error. It involves calculating the error between the detected and the corresponding re-projected feature points. However, this metric uses the same points that were used to estimate the parameters, which reduces its reliability. We decided to simplify and automate the calibration process using multiple checkerboards in a single image for the above reasons. This method could be used to build calibration stations featuring static arrays of checkerboards arranged in a specific configuration. These could be installed inside factories where vehicles with multiple cameras can be accurately calibrated using a single shot without the intervention of specialized staff and simultaneously reducing possible human error while manipulating the calibration targets. Furthermore, the parameter estimation process takes a few seconds helping to reduce the calibration time of multiple vehicles with multiple cameras.

Given the importance of accurate camera intrinsic parameters for 3D applications, we aim to: (1) define clear guidelines to calibrate monocular cameras accurately; and (2) create a method that allows us to calibrate accurately using a single-shot with a predefined setting with multiple checkerboards. To accomplish this, we employ a realistic simulator to generate, calibrate and evaluate hundreds of combinations to obtain the minimum number of checkerboards, their positions, and rotations that would provide accurate intrinsic parameters. We additionally evaluate the corner detection accuracy, which is an integral part of the calibration process, and often overlooked in other work. To overcome the weakness of the re-projection error metric, we intentionally project virtual 3D points, labeled as Control Points, on the camera field of view edges. We then use these Control Points to verify the distortion correction qualitatively and select the best checkerboard arrangements based on score combinations. Finally, we test the top-performing checkerboard poses to calibrate real cameras and project the point cloud generated by a 3D LiDAR sensor using the estimated intrinsic parameters to validate our calibration guidelines findings. To the best of our knowledge, this is the first work to carry out an in-depth study using simulations to provide an optimized set of guidelines for one-shot calibration.

In summary, the main contributions of this work are as follows:The obtention of the minimum practical number of checkerboard poses to calculate the camera projection parameters for 3D applications accurately.The definition of guidelines for checkerboards’ position and rotation w.r.t. to the camera to estimate accurate camera intrinsic parameters.Validation of this single-image calibration setup in the real world with different cameras, lenses, and focal lengths, thus accelerating the camera calibration process.Release of the source code and the synthetic images to facilitate the practical application and reproduction of these guidelines in the real world.

We organize this paper as follows: Section 2 includes a discussion of the related work. Section 3.1 introduces how we obtained the baseline intrinsic parameters based on a real camera. We used these parameters to generate the ground truth synthetic datasets and evaluate the calibration checkerboard poses. Section 3.2 presents the simulator we used throughout this work, the checkerboard model, the synthetic camera, and their coordinate systems. Section 3.5 explains in detail the metrics and the experiments we carried out to understand the practical limitation of the corner detector and draws guidelines to obtain reliable checkerboard corners. Section 3.6 introduces the metrics and the experiments we used to investigate the effects of the checkerboard’s poses on the camera’s intrinsic parameters. This section also obtains multiple checkerboard setups that show reduced error with respect to the ground truth parameters when using one-shot calibration. Section 4.1 presents the testing and evaluation that we performed to replicate the optimal synthetic setups to calibrate a camera in a real-world setting, and estimate the actual intrinsic parameters. Additionally, we present the validation of these parameters by projecting the point cloud generated by a 3D LiDAR into the rectified image using the estimated camera intrinsic parameters. Finally, Section 6 presents our findings and summarizes the guidelines for accurate camera calibration in a single shot that we obtained through synthetic experiments.

## 2. Related Work

There exists a considerable amount of work dedicated to developing techniques for estimation of camera intrinsic parameters. Notable mentions include the work by Zhang [15], Kannala and Brandt [16], and Heikkila and Sliven [17]. Despite being published more than twenty years ago, these methods provide consistent and reliable results. Moreover, the widely-used open-source computer vision library OpenCV [18] and the proprietary Matlab [19] platform use these methods in their camera calibration toolboxes due to their proven accuracy. More recent approaches use deep learning methods to estimate the camera intrinsic parameters using neural networks trained on large datasets of images with known intrinsic parameters [20,21]. These methods are convenient since they do not require any targets or calibration datasets. Nevertheless, these approaches are still far from matching the accuracy achieved by target-based techniques.

Zhang [15] used synthetic data while testing his calibration method to evaluate resilience against noise. He obtained good results with as few as three checkerboards, without aiming to use the parameters in 3D applications. However, he only used the checkerboard corners to measure the error, which might result in over-fitted parameters. Moreover, he did not consider the error introduced by the corner detection phase.

The work dedicated to the extrinsic calibration of LiDARs, radars, and cameras explicitly states that accurate intrinsic camera parameters are required [2,3,4,22,23,24,25,26,27]. These methods find shared features between the 2D perspective space on the images generated by the camera and the 3D Euclidean space employed by radars and LiDARs. The shared features are then input to an optimizer to estimate the extrinsic parameters (relative position t, and rotation R), which attempts to reduce the projection error of the 3D features ($p_{\mathrm{lidar}}$) while using the given camera intrinsic parameters ($P_{\mathrm{lidar},\mathrm{cam}}$) in the form $p_{\mathrm{cam}}=P_{\mathrm{lidar},\mathrm{cam}}\cdot R\cdot t\cdot p_{\mathrm{lidar}}$. This equation illustrates the importance of having high-quality camera intrinsic parameters in order to obtain accurate sensor extrinsic parameters.

There are a limited number of published studies about verifying estimated camera intrinsic parameters for use in 3D applications. Basso et al. [28] stressed the requirement of accurate intrinsic parameters for 3D applications such as SLAM, and introduced a method for the intrinsic and extrinsic calibration optimizer for short-range time-of-flight (ToF) sensors such as the Microsoft Kinect. Geiger et al. [29] presented a single-shot calibration method for short and long-range LiDARs and cameras. Their approach uses multiple checkerboards in a single frame to accelerate and simplify the data acquisition. However, they did not analyze or demonstrate why these positions are optimal. We extend this research direction to create clear guidelines on how to achieve consistent and accurate intrinsic parameters and introduce validation metrics to verify them.

## 3. Methods

### 3.1. Real-World Camera Baseline

While this paper focuses on the use of a simulator to optimize calibration methodology, instead of using a pure virtual camera, we decided to use a real camera as our baseline. We therefore first needed to calibrate it, obtain the intrinsic parameters, and verify that these are appropriate for 3D applications. To calibrate our camera, we decided to use planar checkerboards with checkered patterns since they are widely available, are low cost, and have established corner detection methods that provide high accuracy [29,30]. To make the simulation closer to reality, we decided to model and simulate the checkerboard used to calibrate our real camera.

Additionally, to calibrate our real baseline camera and the virtual cameras generated by the simulator, we re-implemented the corner detection method presented by Geiger et al. [29], based on Ha’s algorithm [31]. This is due to its simplicity and proven advantage in noisy and blurry environments when compared to the Harris [32], and Shi-Tomasi [33] corner detectors included in OpenCV.

#### 3.1.1. Baseline Calibration

We intrinsically calibrated a 5.4 MP Lucid Vision Labs machine vision camera (TRI054S) paired with an 8 mm focal length Fujinon lens. We used an 800 mm by 600 mm planar checkerboard printed on 4 mm thick aluminum, with an eight by six pattern, and a 100 mm square size. A total of 292 checkerboard poses were used to generate a baseline. We then used the OpenCV [18] camera calibration toolkit based on Zhang’s method [15], and MATLAB’s [34] Adaptive Thresholding to obtain the camera intrinsic parameters (principal point, focal length, axis skew), three radial distortion coefficients, and two tangential distortion coefficients. We projected the point cloud generated by a 3D LiDAR sensor, a Hesai Pandar 64, extrinsically calibrated using the method by Zhou et al. [25] to validate that these parameters are accurate for 3D applications.

### 3.2. Simulation

Having a baseline defined by a real camera calibrated with a checkerboard, we created a 3D model with the help of Blender [35], based on the printed planar checkerboard mentioned in Section 3.1.1. We then converted the checkerboard model for use within the LGSVL (LG Silicon Valley) simulator [36] as a controllable object.

The LGSVL Simulator allows the creation of virtual locations, weather scenarios, obstacles, and one ego-vehicle. Any number of sensors can be attached to the ego-vehicle, such as cameras, LiDARs, and GNSS. With the help of the simulator API, we generated an empty scene with an ego vehicle, one camera with the parameters introduced in Section 3.1.1. We then dynamically generated multiple instances of our checkerboard controllable object as illustrated in Figure 1. The camera simulated by the LGSVL simulator rendered images using the plumb-bob model with the given intrinsic parameters. The simulator API allowed us to save these renders as image files.

### 3.3. Checkerboard Coordinate System

We defined the checkerboard coordinate system to be right-handed. The Z-axis is normal to the checkerboard plane, the X-axis is parallel to the checkerboard’s short side, and the Y-axis is parallel to the long side of the checkerboard. The origin is located at the center of the checkerboard as illustrated in Figure 1.

### 3.4. Simulator Coordinate System

The simulator coordinate system is left-handed. The X-axis faces to the right, the Y-axis points upwards, and the Z-axis is normal to the camera plane and faces forward.

### 3.5. Checkerboard Corner Detector Evaluation

Before starting to simulate multiple checkerboards, we decided to initially evaluate the limits of our re-implemented version of the corner detector (based on Geiger et al. [29], and Guiming and Jidong [30]) inside the simulator. We used this information to decide the pose and distance intervals that will have a higher probability of detecting the checkerboard corners, and therefore produce more accurate intrinsic parameters. This step is of utmost importance since the detected checkerboard corners are the inputs for the optimizer. If they contain significant errors, the estimated parameters will be inaccurate.

#### 3.5.1. Corner Detector Metrics

To experimentally obtain the intervals at which the checkerboard corner detector will fail, we located the checkerboard at the center of the camera’s field of view in the simulated world. We rotated the checkerboard with respect to its *X* (roll, α), *Y* (pitch, β), and *Z* (yaw, γ) axes on a [$-90^{\circ }$, $90^{\circ }$] interval with one-degree steps for the roll and pitch and five degrees steps for the yaw; to obtain the maximum distance at which the detector would fail, we moved the checkerboard away from the camera in 1 m steps, until the detector failed. Additionally, to evaluate the corner detector, we defined the following variables and statistics:*Ground Truth 3D corners*$C_{3\mathrm{Dt}}$ are the *N* three-dimensional points in camera space, where N=u×v, and *u* and *v* are the number of inner rows and columns of the checkerboard, respectively.*Corner RMSE* is calculated between the true 2D corners $C_{2\mathrm{Dt}}$ in distorted images generated by the simulator, and the corners computed when running the corner detection $C_{c}$, and calculated as:
$${Corner}_{RMSE}=\sqrt{\frac{\sum_{i=1}^{N}{\left(C_{2Dt_{i}}-C_{c_{i}}\right)}^{2}}{N}}$$The true 2D corners are obtained by projecting the true 3D corners as: $C_{2\mathrm{Dt}}=P\cdot C_{3\mathrm{Dt}}$, where P is the optimized projection matrix from 3D camera space to 2D image space after applying distortion correction, and defined as: $P=\left(\begin{array}{ccc} c_{x} & 0 & f_{x}\\ 0 & c_{y} & f_{y}\\ 0 & 0 & 0 \end{array}\right)$. Translation and rotation are not required in this case because the corners points in $C_{3\mathrm{Dt}}$ are already in camera space.*Inner Checkerboard Area* is calculated by obtaining the area of the two triangles formed by the corners in the checkerboard. Area calculation would be exact in an undistorted image, but is not precise in a distorted one. For this reason, we use two triangles to estimate the area, since we propose that this produces better results than using a parallelogram.*Checkerboard-Image Plane Angle* is defined as the angle between the checkerboard plane normal (as defined by the corners) and the image plane normal (camera z-axis).

#### 3.5.2. Experiments

**Rolling Experiment.** For this experiment, we positioned the origin of the checkerboard at the same height as the camera origin, and set the checkerboard 4 m away along the Z-axis, then varied the roll angle between 0 and 90 degrees in one-degree steps.

**Pitching Experiment.** In this experiment, we aligned the checkerboard and camera origins, placed the checkerboard 4 m away from the camera, and varied the pitch rotation in the [$0^{\circ }$, $90^{\circ }$] interval with one-degree steps.

**Yaw Experiment.** For this experiment, we aligned the checkerboard and camera origins and positioned the checkerboard to be 4 m away from the camera on the Z-axis. We then rotated the checkerboard w.r.t the checkerboard’s Z-axis between [$-90^{\circ }$, $90^{\circ }$] in five degrees steps.

**Simultaneous Rolling and Pitching Experiment.** In this experiment, we examined the effects of simultaneously varying the pitch and roll on the corner detector. We set the checkerboard origin height to match the camera’s and placed the checkerboard 4 m away from the camera along the Z-axis. Additionally, we fixed the yaw rotation to 53.14 degrees. This angle allowed us to align the checkerboard longer diagonal to the vertical axis; this condition helped us simulate the same circumstances that we would use in an actual camera-3D sensor extrinsic calibration [25]. We then simultaneously varied the roll (α) and pitch (β) over the intervals of [$-80^{\circ }$, $80^{\circ }$] and [$-60^{\circ }$, $60^{\circ }$], respectively.

**Range Experiment.** In this experiment, we set the camera origin and the checkerboard origin to have the same height and initially separated them by 4 m along the Z-axis. To verify the maximum detection distance of the checkerboard corner detector, we moved the checkerboard away from the camera in 1 m steps until it failed. Additionally, once we obtained the checkerboard corner detector failure range, we repeated the experiment focusing on the working area with 0.5 m steps to understand better the detector’s performance.

#### 3.5.3. Results

Figure 2 and Figure 3 show the results of the corner detector experiments in the simulator. From these, we can draw the following guidelines regarding the corner detector:We found that the corner detector peak performance with respect to roll rotation between the camera plane normal and the checkerboard normal is between 0 and 60 degrees, as we present in Figure 2a. Rotations below 70 degrees can obtain reliable corner detections, but we can see that the performance quickly decreases at angles larger than 70 degrees, and the detector completely fails for angles larger than 78 degrees.We observed in Figure 2b that the corner detector performs best between 20 and 60 degrees when varying the pitch angle between the camera plane normal and the checkerboard normal. The performance degrades at angles larger than 60 degrees, until it cannot detect any corners at all after 78 degrees.From Figure 2c, we can appreciate that the corner detector performs best between 20 and 60 degrees when simultaneously varying the pitch and roll between the camera plane normal and the checkerboard normal. Similarly, we see a reduction in accuracy when the rotations surpass 78 degrees.From Figure 3a we can see that the corner detector can detect corners reliably up to 35 m, and trivially confirm that corner detector RMSE increases with distance; however, Figure 3b only shows a pronounced drop after 10 m. With the intention of including multiple checkerboards per frame, we can suggest placing the checkerboards within 10 m from the camera or ensure that the checkerboard’s visible inner area should be at least 20,000 $px^{2}$. This value for the area is resolution independent, so we propose it as a guideline for perspective cameras and lenses that produce a different field of view.

### 3.6. Simulated Calibration Experiments

With the knowledge obtained about the impacts of distance and rotation on the corner detector, we investigated the checkerboard positions in the image frame and their influence on the camera intrinsic parameters. To achieve this, we needed to define the metrics to assist evaluation for each set of checkerboard poses and positions.

#### 3.6.1. Checkerboard Pose Metrics

To verify if a set of checkerboard poses provides a better estimation of the camera’s intrinsic parameters, we calculated the root mean square error (RMSE) between the ground truth parameters and the those estimated from the corners of the checkerboards detected on the image using OpenCV [18]. We measured the following parameters:Focal length ($f_{x}$,$f_{y}$).Center point ($c_{x}$,$c_{y}$).Distortion coefficients: three radial ($k_{1}$,$k_{2}$,$k_{3}$), and two tangential ($p_{1}$,$p_{2}$).

In addition to the intrinsic parameters, we also obtained:The RMSE between the ground truth corner positions and the projected corner points using the estimated intrinsic parameters.The checkerboard corner re-projection error, which is the distance between the detected corners in a calibration image, and the corresponding 3D corner points projected into the same image.The Control Points re-projection error, which is the distance between the projections of a 3D Control Point when using the estimated and the ground truth intrinsic parameters. In Section 3.6.2, we introduce and describe the “Control Points” in more detail.

#### 3.6.2. Control Points

The re-projection error is a metric used to quantify the distance between the projection estimate of a 3D point and its actual projection. This metric is widely used to estimate the performance of the camera intrinsic parameters, measuring the detected corners and their 3D estimated counterparts. However, since these points only contain areas belonging to the checkerboard, this metric is unreliable on other parts of the image. For this reason, we decided to purposely insert 3D virtual points into the simulation to assist in measuring the performance on the edges of the image.

The Control Points are 3D virtual points strategically positioned in the camera frustum that we defined in Section 3.1. They target the areas of the camera field of view that are prone to project points incorrectly due to lens distortions. We located these points systematically as shown in Figure 4 at 5 m and 50 m from the camera origin. We defined those two distances to test the performance at close and long-range.

Figure 5 shows two simulated checkerboards with an almost identical re-projection error value when using only the checkerboard corners. The ground truth checkerboard corners are drawn with a red crosshair, while the re-projected corners are drawn with a green crosshair. The corner points for both checkerboards are re-projected after un-distortion with sub-pixel accuracy. However, when using the Control Points to calculate the re-projection error in Figure 5a, the error metric increases considerably. The checkerboard in Figure 5b, on the other hand, has lower control point re-projection error due to the intrinsic parameters being closer to the ground truth. We use this new metric to help us determine whether a checkerboard pose is adequate or not.

#### 3.6.3. Dual Checkerboard Calibration

In this series of experiments, we evaluated the effect of varying the position and rotation of each checkerboard. This paper aims to formulate guidelines that will help to narrow down the number of combinations required when increasing the number of checkerboards. To do this, we must determine the poses that result in a minimized error.

Having explained the proposed method for corner detector evaluation and defined the required metrics, we also aim to evaluate how the following items affect the calibration parameters:The rotation angles (Section 3.6.4).The vertical and horizontal checkerboard positioning (Section 3.6.5 and Section 3.6.6).The distance between the camera and the checkerboard (Section 3.6.7).The number of checkerboards or poses (Section 3.6.9).

Finally, we present the results of all the above items in Section 3.6.8.

#### 3.6.4. Dual Checkerboard Rotation Experiments

In these experiments, we investigate the checkerboard pose performance when varying the roll (α) and pitch (β) angles in different rotational combination patterns: individual, simultaneous, symmetric, and asymmetric. For all these experiments, we fixed the checkerboard yaw angle to 53.14 degrees because this aligns the checkerboard longer diagonal to the vertical-axis, which allowed us to simplify the camera-LiDAR calibration [25]. This condition was required to simulate the same settings to use in the real-world scenario.

We used the roll and pitch rotation intervals we obtained in Section 3.5 for the following experiments, since we previously determined that these would provide accurate checkerboard corner detections.

We prepend all of the experiments in this section by the letter **A**, denoting the angle variation. The following list summarizes the experiments we carried out, with the rotation experiments being additionally summarized in Table 1:

**A1**. Varying the pitch and roll of the right checkerboard, while keeping the left checkerboard static. We simulated all the roll (α) and pitch (β) combinations over the [$-60^{\circ }$, $60^{\circ }$] interval in 10-degree steps.

**A2**. Varying the pitch and roll of the left checkerboard, while keeping the right checkerboard static. We varied the roll and pitch over the [$-60^{\circ }$, $60^{\circ }$] interval in 10-degree steps.

**A3**. Varying the pitch and roll of the left checkerboard and right checkerboard simultaneously over the [$-60^{\circ }$, $60^{\circ }$] interval in 10-degree steps. The yaw angle of both checkerboards was set to −53.14 degrees.

**A4**. This experiment was similar to A3. It varied the pitch and roll of the left checkerboard and right checkerboard simultaneously over the [$-60{\;}^{\circ }$, $60^{\circ }$] interval in 10-degree steps. However, in this experiment, we mirrored the yaw angle between checkerboards, setting the left $\gamma_{l}$ angle to 53.14, and the right $\gamma_{l}$ angle to −53.14 degrees.

**A5**. Similar to the A3 experiment, we varied simultaneously the roll and pitch angles in the [$-60^{\circ }$, $60^{\circ }$] interval, in 10 -degree steps. But in this experiment, we mirrored all the rotations ($\alpha_{l}=-\alpha_{r}$, $\beta_{l}=-\beta_{r}$, and $\gamma_{l}=-\gamma_{r}$).

#### 3.6.5. Dual Checkerboard Horizontal Positioning Experiments

In these experiments, we investigated the checkerboard pose performance when varying the horizontal positioning for two simultaneous checkerboards in the camera field-of-view. We used different combinations while keeping the vertical positioning fixed.

From Section 3.5.3, we obtained the performance limits for the object detector we selected on Section 3.2. Additionally, we confirmed that the closer the checkerboard is to the camera plane, the better the checkerboard corner detector performance. For this reason, we positioned both checkerboards five meters from the camera plane along the camera’s Z-axis. Additionally, for these experiments we selected the roll and pitch rotations that provided the best corner detection results. With these initial conditions, we simulated the following experiments involving variation of the horizontal positioning of the checkerboards. We prepend all of the experiments in this section by the letter **H**, denoting the horizontal variation. We also summarize the experiment parameters and results in Table 2:

**H1**. In this experiment, we modified the horizontal position of both checkerboards simultaneously along the camera X-axis, while each checkerboard was facing the camera with different rotation angles. We positioned the left checkerboard at the left edge of the image, with the right checkerboard next to its right side separated by 0.9 m. Additionally, we set the right checkerboard to have −40, −20, and −53.14 degrees for the roll, pitch, and yaw, respectively; and 0, 0, and −53.14 degrees for the left checkerboard. We then moved both checkerboards with 0.02 m steps until the right checkerboard reached the image edge.

**H2**. Similar to H1, in this experiment we changed only the horizontal position until the right checkerboard reached the image edge, in 0.02 m steps. However, the checkerboard normals had the rotation angles mirrored. The left checkerboard roll, pitch, yaw rotations were set to 50, −20, and −53.14 degrees, respectively, while the right ones were set to −50, 20, and −53.14 degrees.

**H3**. Contrary to the H1 and H2, in this experiment we fixed the initial position of both checkerboards to the center of the image and moved the horizontal position of each checkerboard simultaneously towards the left and right image edges in 0.01 m steps. We set the right checkerboard roll, pitch, and yaw rotations to −40, −20, and −53.14 degrees, respectively, while setting the left checkerboard rotations to 0, 0, and −53.14 degrees.

**H4**. In this experiment, we initially positioned both checkerboards at the center of the frame, separated by 0.8 m. We then moved each checkerboard towards the left and right edges, respectively, in 0.01 m steps. We also set the left and right checkerboards to point towards opposite directions. The right checkerboard roll was set to −50, the pitch to 20, and the yaw to −53.14 degrees; the left one was set to 50, −20, and −53.14 degrees in roll, pitch, and yaw, respectively.

**H5**. In this experiment, we set both checkerboards at the center of the frame separated by 0.8 m, with their rotation angles mirrored. We then moved the left checkerboard towards the left edge of the image and simultaneously moved the right checkerboard to the right side of the frame. Both checkerboards were moved in 0.01 m steps. The right checkerboard roll, pitch, and yaw rotations were set to 50, −20, and −53.14 degrees, respectively, while the left checkerboard rotations were set to −50, 20, and −53.14 degrees.

#### 3.6.6. Dual Checkerboard Vertical Positioning Experiments

In these experiments, we examined the checkerboard pose performance when varying symmetrically and asymmetrically the vertical positioning of the two checkerboards in the camera field-of-view. Similar to the experiments of Section 3.6.5, we selected the most performant rotations for the checkerboard detector from Section 3.5.3. The following list explains the vertical positioning experiments. We prepend the experiments in this section with **V**, denoting the vertical variation. We additionally include a summary of these experiments in Table 2:

**V1**. In this experiment, we initially positioned the left checkerboard at the bottom left edge and the right checkerboard at the lower right of the frame. We fixed the rotations of both checkerboards to −40, −20, and −53.14 degrees for the roll, pitch, and yaw, respectively. We then moved both checkerboards simultaneously to the top of the image in steps of 0.02 m, without modifying the horizontal position.

**V2**. In this experiment, we initially positioned the left checkerboard to the top-left edge and the right checkerboard to the lower right of the frame. We fixed both checkerboard rotations to −40, −20, and −53.14 degrees for the roll, pitch, and yaw, respectively. We then moved the left checkerboard towards the bottom left and the right checkerboard to the top right in steps of 0.02 m, without modifying the horizontal position.

#### 3.6.7. Dual Checkerboard Distance Experiments

Although we know that the corner detector performs better the closer the checkerboard is to the camera, in these experiments we evaluated the impacts of checkerboard scale on the accuracy of the intrinsic parameters. Similar to the horizontal and vertical experiments, we selected the most performant poses from Section 3.5.3. Additionally, from Section 3.6.5 we chose to keep the checkerboards near the left and right edges since this positioning provided better parameter estimation. The following list describes the distance experiments. We prepend the experiments in this section with **D**, denoting the distance. We additionally include a summary of these experiments in Table 3:

**D1**. In this experiment, we initially positioned both checkerboards at the camera height, but distanced the left checkerboard 5 m from the camera and the right one at 10 m. We kept the left checkerboard static during the whole experiment on the left image edge, while we reduced the right checkerboard distance in 0.1 m steps until it reached 4.6 m. The roll, pitch, and yaw for both checkerboards was set to −40, −20, and −53.14 degrees, respectively.

**D2**. In this experiment, we initially positioned both checkerboards at the camera height, with one on the left image edge and the other on the right. We set the initial distance of both checkerboards to 10 m, and then moved them towards the camera until they reached a distance of 5 m from the camera. We fixed the roll, pitch, and yaw for both checkerboards to −40, −20, and −53.14 degrees, respectively.

#### 3.6.8. Dual Checkerboard Calibration Results

Looking at the results from Table 1, Table 2 and Table 3, we can infer the following guidelines:1.It is essential to locate the checkerboards near the image edge in order to generate distortion parameters closer to the ground truth, a well-known fact [11,37,38].2.Having a single checkerboard pointing directly to the camera while the other is rotated seems to produce better overall parameters. We hypothesize that this dual checkerboard combination provides enough information to estimate the center point and the focal length accurately, while also producing sufficient data to calculate the distortion parameters.3.Keeping the checkerboard rotation inside the [$-60^{\circ }$, $60^{\circ }$] interval, as we defined in Section 3.5.3, is essential to increase the corner detection accuracy.4.Checkerboards located on the image corners are not required, as long as the user can position the checkerboards near the image edge.5.Checkerboards with varied rotation relative to the camera plane are more beneficial than those located farther away, since closer checkerboards produce more accurate corners. We found this to be true because rotated checkerboards near the acceptable rotation limits also provide scale variations.

#### 3.6.9. Multiple Checkerboards Calibration

Based on the guidelines we defined in Section 3.6.8, we defined the poses of up to 10 checkerboards in the image frame. Since we narrowed down the combinations, all these experiments contain only fixed positions and rotations, instead of simulation intervals. Additionally, for the top-score poses, we manually picked other performant poses to form two setups: one with six and the other with seven checkerboards. To form these “custom” setups, denoted by subscript **C**, we chose what we considered “easy” to replicate positions in the real world. For instance, we preferred (but did not prioritize) checkerboard positions closer to the ground or separated from each other; this would help accelerate checkerboard positioning setup in a real-world scenario. However, we always followed the guidelines we defined in Section 3.6.8.

The summary of the simulation experiments with multiple checkerboards is in Table 4. We decided to limit the number of checkerboards to 10 because adding more checkerboards would prove a challenge to set up in the real and simulated worlds while maintaining all the checkerboards at a close range to maximize the performance of the corner detector.

#### 3.6.10. Multiple Checkerboards Calibration Results

Table 5 and Table 6 show that it might be feasible to calibrate with as few as three checkerboards and obtain good calibration results. However, in Figure 6 we can see that the $\alpha_{1}$ experiment, with three checkerboards, presents a larger than average focal length error, which might produce scaling problems and an irregular projection at different distances.

Additionally, on the same Section 3.6.10, we can see that having more checkerboards produces more accurate calibration intrinsic parameters. However, we can also infer that using six or seven checkerboards can provide enough information to estimate highly accurate calibration parameters for 3D applications. This number of checkerboards gives a realistic and balanced approach since having more than seven simultaneous checkerboards in the frame might cause difficulties while replicating the setups presented in Table 4 when using real checkerboards and tripods. For this reason, in the next section, we decided to investigate and test these setups with real cameras, project the point cloud from a 3D LiDAR, and verify the quality of the camera intrinsic parameters.

## 4. Results

### 4.1. Real-World Calibration Verification

For real-world calibration verification, we used the Lucid Labs machine vision camera that was introduced in Section 3.1.1, and an additional FLIR Blackfly camera (PGE-23S6C), paired with an 8 mm μTron lens. This camera has a lower resolution (1920 × 1200 pixels, approximately 2.3 MP) and a larger image sensor, leading to a wider field of view even if it uses a lens with the same focal length. Additionally, we paired a 25 mm Fujinon lens with the Lucid camera to verify a telephoto scenario. We calibrated and validated the three sets as described in Section 3.1.1. A summary of the cameras we used to validate our guidelines is in Table 6.

These experiments would also help us study how our guidelines perform on multiple camera resolutions and wide and narrow fields of view. Wide-angle lenses are commonly used in autonomous driving applications or robotics for peripheral sensing. In contrast, telephoto lenses are used for long-range applications, such as traffic light detection and classification, or object detection at long-range when running on highways. We proceeded to replicate the checkerboard positions we obtained in Section 3.6.10 for the setups with six and seven checkerboards: $\delta_{1},\delta_{C},\epsilon_{1},\epsilon_{C}$, for the 8 mm and 25 mm Fujinon lenses on the Lucid and the 8 mm μTron lens on the FLIR camera, a total of 12 experiments.

### 4.2. Multiple Checkerboard Verification Experiments

The checkerboard poses we obtained from the simulator are defined by their roll, pitch, and yaw rotations. These angles are numbers that we, as humans, might find difficult to replicate in the real world with such accuracy. For this reason, and to simplify and accelerate the checkerboard positioning in a garage, we created transparent image overlays of the checkerboard poses rendered by the simulator. With the help of the Robot Operating Systems (ROS) [39], we wrote a node that receives the camera stream, superimposes in real-time the overlay, and projected the composed image on a large screen while replicating the checkerboard poses defined by the $\delta_{1},\delta_{C},\epsilon_{1},\epsilon_{C}$ experiments in a garage. Figure 7 shows the setup we used while matching the checkerboards poses with the help of tripods. To simplify the replication of our experiments by the community, we publicly released the image overlays and the ROS node to the following repository: https://gitlab.com/perceptionengine/pe-datatools/ros_image_overlay (accessed on 29 January 2022).

Having matched the checkerboard positions with our simulation setups, we captured a single image for the $\delta_{1}$, $\delta_{c}$, $\epsilon_{1}$ and $\epsilon_{c}$ experiments from Section 3.6.9 with the Lucid and the FLIR cameras; we then proceeded to detect the checkerboard corners with the corner detector introduced in Section 3.1.1, and obtained the intrinsic parameters for each camera setup using OpenCV [18].

In addition to the one-shot calibration dataset, we captured camera and LiDAR data outdoors. With the help of this dataset, we obtained the extrinsic calibration parameters between the Lucid and FLIR cameras and the 3D LiDAR, which we summarized in Table 7. We then individually projected the 3D LiDAR point cloud into the same image captured by the camera using the four sets of intrinsic parameters ($\delta_{1},\delta_{C},\epsilon_{1},\epsilon_{C}$). We repeated this procedure for the lower resolution FLIR camera, and the Lucid with the telephoto lens.

## 5. Discussion

### Real-World Calibration Results

Having estimated the intrinsic parameters for the setups outlined in Table 8 and Table 9 using the single-shot setups, we obtained the absolute error between these intrinsic parameters and our baseline values. We prepared qualitative tests, for which we projected the point cloud from a Hesai Pandar 64 LiDAR on the rectified image by each set of estimated parameters from the $\delta_{1},\delta_{C},\epsilon_{1},\epsilon_{C}$ experiments and compiled them in Figure 8, Figure 9 and Figure 10. Additionally, as a quantative evaluation, we calculated the checkerboard corner re-projection error, and summarized the results in Figure 11. In this figure, we can verify that both of the seven checkerboard setups provided the most reliable intrinsic parameters, as we expected.

Both of the ϵ experiments (seven checkerboards) show better overall performance when compared with their equivalent δ counterparts (six checkerboards), as can be observed in Figure 11a,b. This holds true for both the Lucid cameras and the FLIR camera, which is consistent with our simulation results in Figure 6 and the qualitative results in Figure 8 and Figure 9. The best performing experiment in our simulation ($\epsilon_{C}$) obtained the lowest error in our real-world experiments. Moreover, the corresponding qualitative results from both cameras also exhibit an excellent point cloud projection as illustrated by Figure 9g,h and Figure 10g,j.

The worst performing simulated experiment ($\delta_{C}$) also matched with our real experiment results, in both the wide angle and the telephoto lens experiments. While obtaining the extrinsic parameters, we noted that the roll rotation for both cameras in the $\delta_{C}$ test is slightly different from the rest of the experiments, as we can see in Table 7. The extrinsic calibration method searches for shared features in the image and LiDAR domain and uses these as an input for an optimizer to minimize the re-projection error. It utilizes the detected 3D features and projects them using the given camera intrinsic parameters. However, if the intrinsic parameters erratically project the 3D features, the optimizer will estimate an incorrect relative camera-LiDAR position, leading us to infer that the camera-LiDAR extrinsic calibration algorithm incorrectly converged due to inaccurate intrinsic parameters. Additionally, after analyzing the qualitative results for the $\delta_{C}$ experiment, we noticed an adequate projection at long range, but on the right-bottom section of Figure 8d,c and Figure 9d,c we found that the point cloud hitting the barricade located closer to the camera is incorrectly projected, pointing out a large error in the focal length. This can be confirmed in the quantitative results presented in Figure 11a,b.

Telephoto lenses tend to expose a low pincushion distortion in the center of the image, which is characterized by larger $k_{3}$ coefficients. For this reason, it might appear that the experiment $\epsilon_{1}$ with the telephoto lens is performing worse in terms of the distortion coefficients. However, even if the difference we appreciate in Figure 11c might seem significant, that difference only slightly affects the edges of the image as shown in Figure 12; qualitative results in Figure 10 also show that ϵ parameters provide a good point cloud projection. Additionally, in terms of extrinsic calibration, we see that similar to the wide-angle experiments, close-range projection on the δ experiments appear to have incorrect size as shown in the barricade we purposely put close to the camera (Figure 10a,c). Finally, it is essential to mention that we carried out the experiments with the telephoto camera using a slightly different setup, for that reason the values of the extrinsic parameters for the telephoto setup in Table 7 are different from its wide-angle counterparts.

## 6. Conclusions

We presented a first-of-its-kind method to generate clear guidelines for single shot camera intrinsic calibration using multiple checkerboards, suitable for use in 3D applications. With the help of a simulator, we defined the position and rotation intervals that allow a corner detector to obtain optimal detections; we then generated thousands of multiple checkerboard poses and evaluated them to obtain position and rotation intervals that maximize the probability of estimating accurate camera intrinsic parameters. These results gave us enough information to generate checkerboard pose guidelines. Using these guidelines, we developed sets of multiple checkerboard poses and evaluated them synthetically and in the real world using three different camera sets with different resolutions and fields of view.

The overall results show that the camera simulations helped us to accelerate the camera modeling process, its evaluation, and ultimately the creation of guidelines to obtain accurate intrinsic parameters. We can also infer that even if the simulations create ideal image conditions, i.e., images without chromatic aberration, vignetting, and so on, we can still transfer the lessons learned to the real world. It would have been challenging and costly to exactly replicate the simulated experiments in the real world since they require specialized equipment to position and rotate the checkerboards. Moreover, to obtain the ground truth corner coordinates, a team of labelers would be necessary to identify each corner at the pixel level, extending the time required to complete this work.

In this paper, we simulated and evaluated a single plumb-bob camera synthetically. For this reason, the distances we mentioned might not apply to other focal length lenses. Nevertheless, as an additional finding, we learned in Section 3.5.3 that if the checkerboard guidelines we defined in Section 3.6.10 are to be used with a different field of view camera, instead of following the recommended checkerboard distance, the user should aim to project the checkerboard area with at least 20,000 $px^{2}$ to produce precise corner detection. We validated this experimentally when testing on the FLIR camera in Section 4.1, which has a wider field of view and less than half the resolution of the Lucid camera. We additionally validated our guidelines on a telephoto lens, with about a third of the field-of-view of the simulated lens. It is important to note that the simulated experiments and control point validation were designed for a specific wide-angle lens and should be different for a telephoto lens with a narrower field of view and longer reach. Furthermore, our results might not be applied to cameras paired with fish-eye lenses since these are normally modeled with non-perspective projection models, which we aim to test in future work. Nevertheless, we found consistent results between our simulations and our real-world evaluations thanks to the real-world experiments and the validation experiments with the telephoto lens. This enabled us to confirm that if our guidelines are followed, accurate intrinsic parameters for 3D applications can be obtained by using seven checkerboards.

Finally, the checkerboard pose guidelines we defined in Section 3.6.8 and Table 4 can also be used with the original method presented by Zhang [15], which involves using a single checkerboard and capturing multiple images with random orientations. Instead of moving the checkerboard randomly around the whole camera field of view, the user might aim to replicate the checkerboard poses we defined for the $\delta_{1}$, $\epsilon_{C}$ experiments, or even the ζ, η or θ experiments, and use the images as an input for either the OpenCV or Matlab calibration toolboxes. This would simplify and accelerate the calibration process, while helping to estimate accurate intrinsic parameters for 3D applications such as robotics, autonomous driving, or other applications that require high quality parameters.

## Figures and Tables

**Figure 1 sensors-22-02067-f001:**
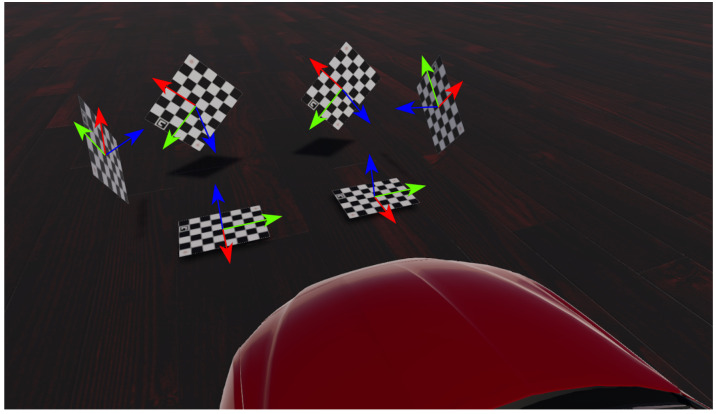
Our modeled checkerboard simulated in the LGSVL.

**Figure 2 sensors-22-02067-f002:**
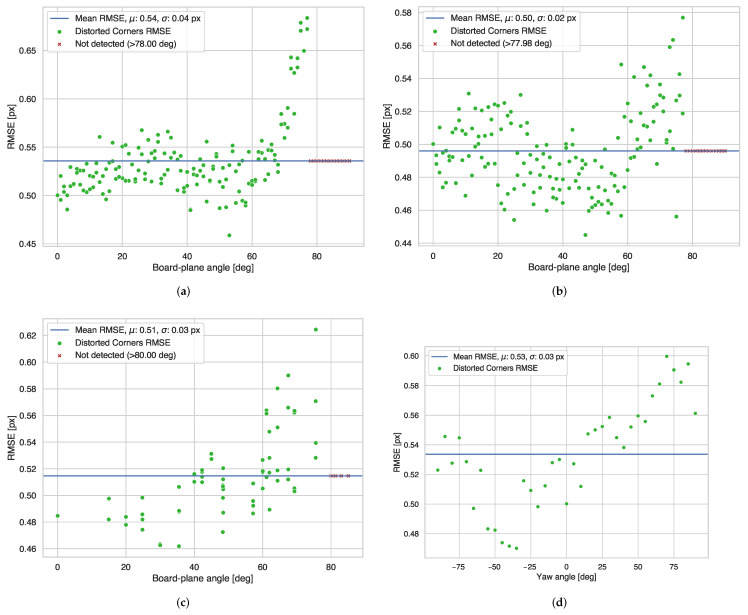
Effects of roll (α), pitch (β) and yaw (γ) rotations on the checkerboard corner detector. (**a**) Effects of Roll (α) Angle, (**b**) Effects of Pitch (β) Angle, (**c**) Effects of Simultaneous Roll (α) and Pitch (β), and (**d**) Effects of Yaw (γ) Angle.

**Figure 3 sensors-22-02067-f003:**
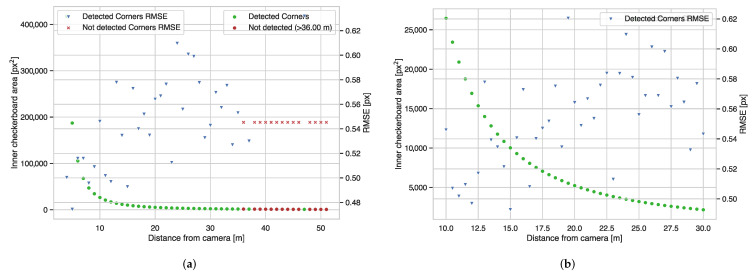
Effects of distance between the checkerboard and the camera on the checkerboard corner detector. (**a**) Effects of Distance, 4 to 51 m interval, (**b**) Effects of Pitch (β) Angle.

**Figure 4 sensors-22-02067-f004:**
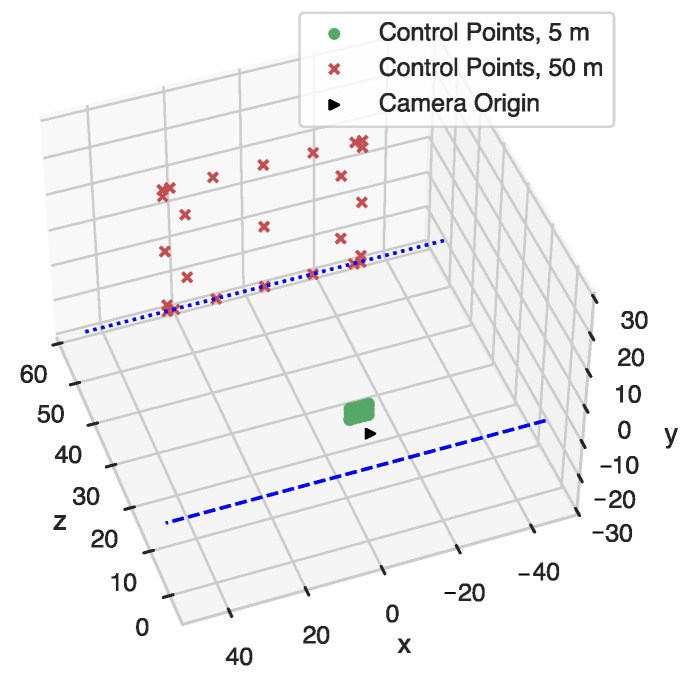
Control Points systematically located inside the baseline camera frustum.

**Figure 5 sensors-22-02067-f005:**
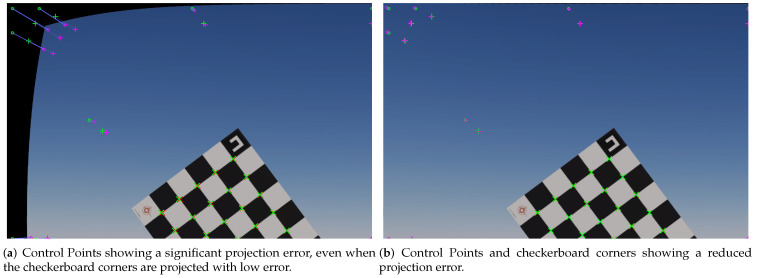
Control points as an auxiliary metric. Green marks represent the “Control points” projected with the ground truth intrinsic parameters, while purple marks represent the projection of the “Control Points” using the estimated intrinsic parameters. Both (**a**,**b**) have the same subpixel checkerboard corner re-projection error value, and corners in the checkerboard are correctly re-projected in both cases. However, the estimated intrinsic parameters have a large error in (**a**), correlating to Control Point re-projection error.

**Figure 6 sensors-22-02067-f006:**
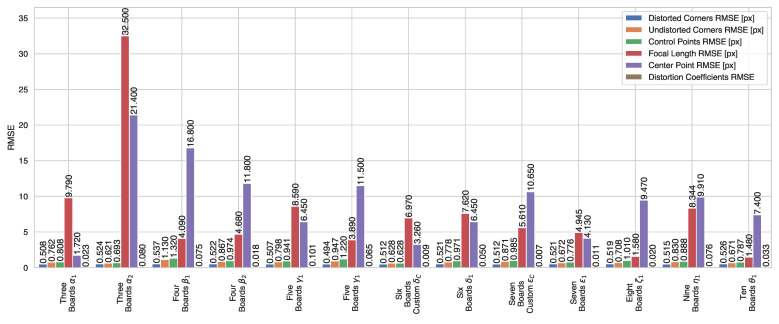
Quantitative result summary for the simulation experiments with multiple checkerboards.

**Figure 7 sensors-22-02067-f007:**
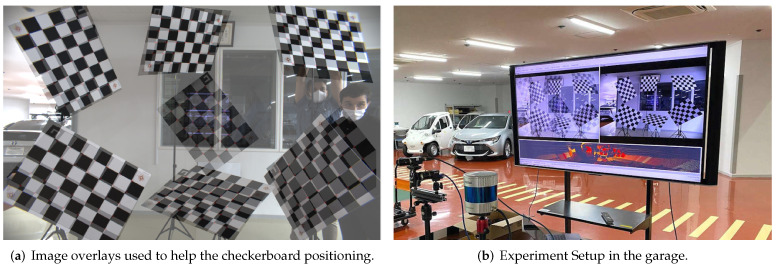
Multiple checkerboard verification experiments in a garage. (**a**) shows the composed image by the camera stream and the checkerboards overlays. (**b**) shows the camera and the screen we used during the experiments to help us match the checkerboards poses.

**Figure 8 sensors-22-02067-f008:**
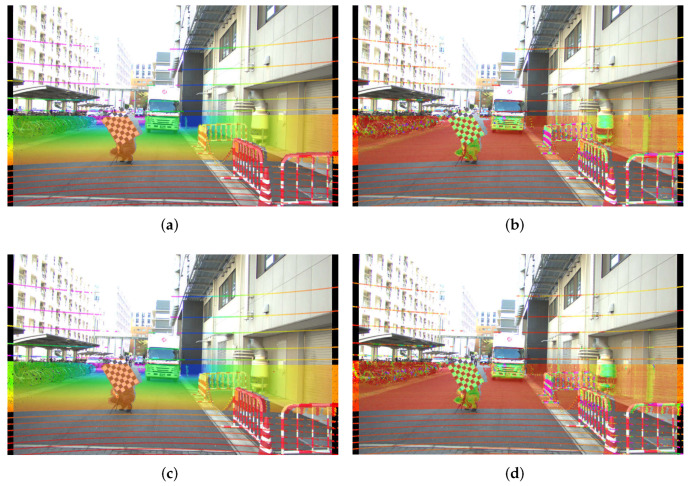
Point cloud projection on the Lucid camera with the wide angle lens, using the intrinsic parameters by the one-shot experiments, replicating the $\delta_{1},\delta_{C},\epsilon_{1},and\epsilon_{C}$ experiments. In the left column, the projected point cloud is colored by distance. In the right column, the projected point cloud is colored by the laser intensity of each return value. (**a**) Projection using $\delta_{1}$’s intrinsic parameters, point cloud colored by depth, (**b**) projection using $\delta_{1}$’s intrinsic parameters, point cloud colored by laser intensity, (**c**) projection using $\delta_{C}$’s intrinsic parameters, point cloud colored by depth, (**d**) projection using $\delta_{C}$’s intrinsic parameters, point cloud colored by laser intensity, (**e**) projection using $\epsilon_{1}$’s intrinsic parameters, point cloud colored by depth, (**f**) projection using $\epsilon_{1}$’s intrinsic parameters, point cloud colored by laser intensity, (**g**) projection using $\epsilon_{C}$’s intrinsic parameters, point cloud colored by depth, and (**h**) projection using $\epsilon_{C}$’s intrinsic parameters, point cloud colored by laser intensity.

**Figure 9 sensors-22-02067-f009:**
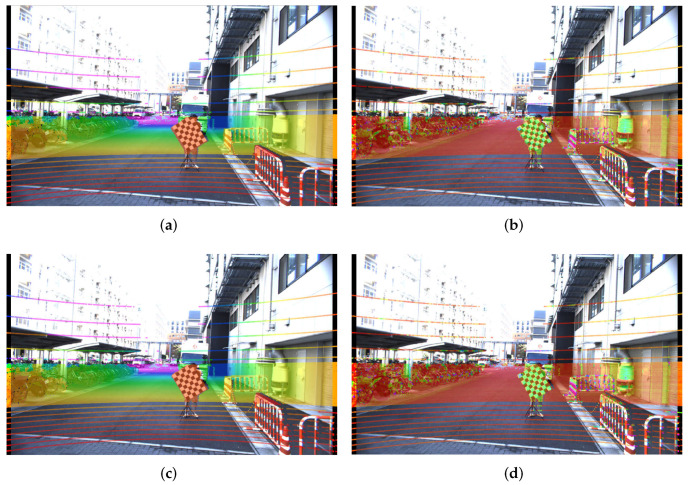
Point cloud projection on the FLIR camera with the wide angle lens, using the intrinsic parameters by the one-shot experiments, replicating the $\delta_{1},\delta_{C},\epsilon_{1},\epsilon_{C}$ experiments. Figures on the left column colored the projected point cloud by distance. In the left column, the projected point cloud is colored by distance. In the right column, the projected point cloud is colored by the laser intensity of each return value. (**a**) Projection using $\delta_{1}$’s intrinsic parameters, point cloud colored by depth, (**b**) projection using $\delta_{1}$’s intrinsic parameters, point cloud colored by laser intensity, (**c**) projection using $\delta_{C}$’s intrinsic parameters, point cloud colored by depth, (**d**) projection using $\delta_{C}$’s intrinsic parameters, point cloud colored by laser intensity, (**e**) projection using $\epsilon_{1}$’s intrinsic parameters, point cloud colored by depth, (**f**) projection using $\epsilon_{1}$’s intrinsic parameters, point cloud colored by laser intensity, (**g**) projection using $\epsilon_{C}$’s intrinsic parameters, point cloud colored by depth, and (**h**) projection using $\epsilon_{C}$’s intrinsic parameters, point cloud colored by laser intensity.

**Figure 10 sensors-22-02067-f010:**
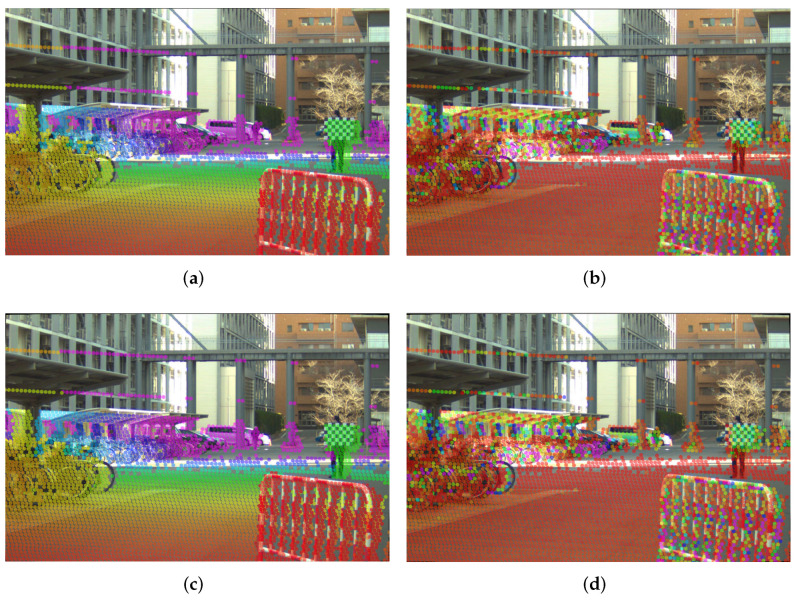
Point cloud projection on the Lucid camera with the Telephoto lens, using the intrinsic parameters by the one-shot experiments, replicating the $\delta_{1},\delta_{C},\epsilon_{1},and\epsilon_{C}$ experiments. In the left column, the projected point cloud is colored by distance. In the right column, the projected point cloud is colored by the laser intensity of each return value. (**a**) Projection using $\delta_{1}$’s intrinsic parameters, point cloud colored by depth, (**b**) projection using $\delta_{1}$’s intrinsic parameters, point cloud colored by laser intensity, (**c**) projection using $\delta_{C}$’s intrinsic parameters, point cloud colored by depth, (**d**) projection using $\delta_{C}$’s intrinsic parameters, point cloud colored by laser intensity, (**e**) projection using $\epsilon_{1}$’s intrinsic parameters, point cloud colored by depth, (**f**) projection using $\epsilon_{1}$’s intrinsic parameters, point cloud colored by laser intensity, (**g**) projection using $\epsilon_{C}$’s intrinsic parameters, point cloud colored by depth, and (**h**) projection using $\epsilon_{C}$’s intrinsic parameters, point cloud colored by laser intensity.

**Figure 11 sensors-22-02067-f011:**
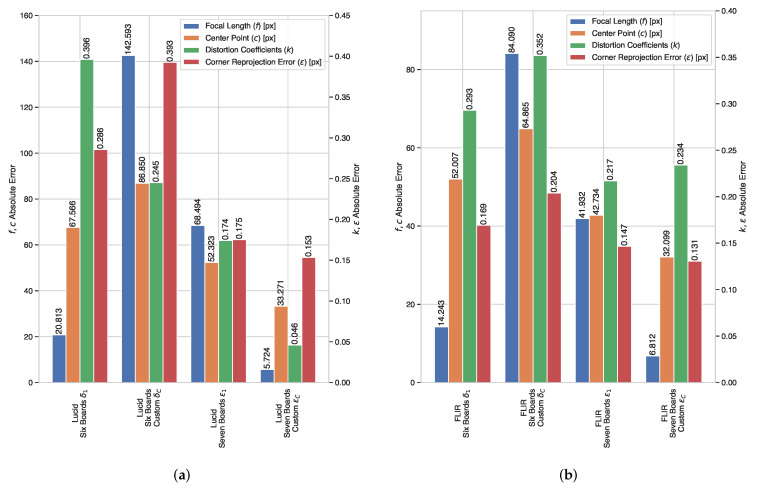
Real-world results for the multi-checkerboard experiments. The left axis denotes the error in pixels for the sum of both focal lengths (*f*) and the sum of the center point (*c*). The right axis denotes the absolute error for the sum of distortion coefficients ($k_{1},k_{2},k_{3},p_{1},p_{2}$) and checkerboard corner re-projection error (ϵ). (**a**) Absolute error comparison for the Lucid camera with the wide angle lens, (**b**) absolute error comparison for the FLIR camera with the wide angle lens, and (**c**) absolute error comparison for the Lucid camera with the telephoto lens.

**Figure 12 sensors-22-02067-f012:**
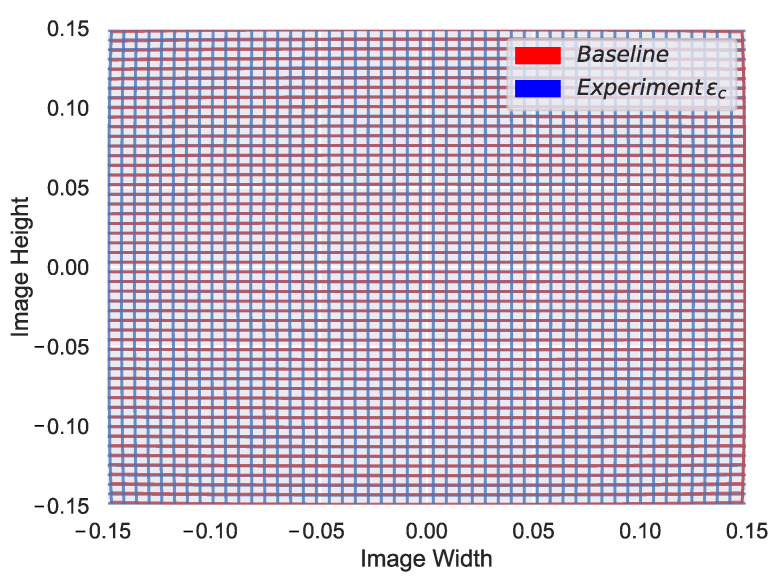
Third-order radial distortion coefficient effect comparison on the telephoto lens.

**Table 1 sensors-22-02067-t001:** Summary of the rotation experiments with dual checkerboards and their results. The best poses are defined by the top 2% lowest error of the projected control points. The subscript letter identifies the left (${}_{l}$) and right (${}_{r}$) checkerboards, respectively. The # symbol in parenthesis represents the index of the experiment.

Experiment	Left Checkerboard Parameters	Right Checkerboard Parameters	Top 2% Poses, Its Control Points, and Corners Re-Projection Error ($\epsilon_{\mathrm{ctrl}},\epsilon_{\mathrm{corner}}$) [px]
**A1** Left Fixed Right Rotate	x,y,z: (−0.6, 1.0, 5.0) m α: 0° β: 0° γ: −53.14°	x,y,z: (0.6, 1.0, 5.0) m α: [−60, 60, 10]° β: [−60, 60, 10]° γ: −53.14°	(#98) $\alpha_{r}$: 10°, $\beta_{r}$: 10°, $\epsilon_{ctrl}$: 1.807, $\epsilon_{corner}$: 0.1543 (#72) $\alpha_{r}$: −10°, $\beta_{r}$: 10°, $\epsilon_{ctrl}$: 1.8260, $\epsilon_{corner}$: 0.1484 (#110) $\alpha_{r}$: 20°, $\beta_{r}$: 0°, $\epsilon_{ctrl}$: 1.8867, $\epsilon_{corner}$: 0.1611 (#30) $\alpha_{r}$: −40°, $\beta_{r}$: −20°, $\epsilon_{ctrl}$: 1.8930, $\epsilon_{corner}$: 0.1605
**A2** Left Rotate Right Fixed	x,y,z: (−0.6, 1.0, 5.0) m α: [−60, 60, 10]° β: [−60, 60, 10]° γ: −53.14°	x,y,z: (0.6, 1.0, 5.0) m α: 0° β: 0° γ: −53.14°	(#52) $\alpha_{l}$: −20°, $\beta_{l}$: −60°, $\epsilon_{ctrl}$: 1.1576, $\epsilon_{corner}$: 0.1686 (#132) $\alpha_{l}$: 40°, $\beta_{l}$: −40°, $\epsilon_{ctrl}$: 1.2514, $\epsilon_{corner}$: 0.1685 (#94) $\alpha_{l}$: 10°, $\beta_{l}$: −30°, $\epsilon_{ctrl}$: 1.3005, $\epsilon_{corner}$: 0.1674 (#92) $\alpha_{l}$: 10°, $\beta_{l}$: −50°, $\epsilon_{ctrl}$: 1.3379, $\epsilon_{corner}$: 0.1507
**A3** Left Rotate Right Rotate	x,y,z: (−0.6, 1.0, 5.0) m α: [−60, 60, 10]° β: [−60, 60, 10]° γ: −53.14°	x,y,z: (0.6, 1.0, 5.0) m α: [−60, 60, 10]° β: [−60, 60, 10]° γ: −53.14°	(#132) $\alpha_{l}$: 40°, $\beta_{l}$: −40°, $\alpha_{r}$: 40°, $\beta_{r}$: −40°, $\epsilon_{ctrl}$: 1.5329, $\epsilon_{corner}$: 0.1915 (#58) $\alpha_{l}$: −20°, $\beta_{l}$: 0°, $\alpha_{r}$: −20°, $\beta_{r}$: 0°, $\epsilon_{ctrl}$: 1.6313, $\epsilon_{corner}$: 0.1535 (#157) $\alpha_{l}$: 60°, $\beta_{l}$: −50°, $\alpha_{r}$: 60°, $\beta_{r}$: 50°, $\epsilon_{ctrl}$: 1.8264, $\epsilon_{corner}$: 0.1928 (#36) $\alpha_{l}$: −40°, $\beta_{l}$: 40°, $\alpha_{r}$: −40°, $\beta_{r}$: 30°, $\epsilon_{ctrl}$: 1.8902, $\epsilon_{corner}$: 0.1752
**A4** Left Rotate Right Rotate Mirror Yaw	x,y,z: (−0.6, 1.0, 5.0) m α: [−60, 60, 10]° β: [−60, 60, 10]° γ: 53.14°	x,y,z: (0.6, 1.0, 5.0) m α: [−60, 60, 10]° β: [−60, 60, 10]° γ: −53.14°	(#28) $\alpha_{l}$: −40°, $\beta_{l}$: −40°, $\alpha_{r}$: −40°, $\beta_{r}$: −40°, $\epsilon_{ctrl}$: 1.5135, $\epsilon_{corner}$: 0.17 (#98) $\alpha_{l}$: 10°, $\beta_{l}$: 10°, $\alpha_{r}$: 10°, $\beta_{r}$: 10°, $\epsilon_{ctrl}$: 1.6489, $\epsilon_{corner}$: 0.1634 (#73) $\alpha_{l}$: −10°, $\beta_{l}$: 20°, $\alpha_{r}$: −10°, $\beta_{r}$: 20°, $\epsilon_{ctrl}$: 1.9723, $\epsilon_{corner}$: 0.1635 (#68) $\alpha_{l}$: −10°, $\beta_{l}$: −30°, $\alpha_{r}$: −10°, $\beta_{r}$: −30°, $\epsilon_{ctrl}$: 2.1704, $\epsilon_{corner}$: 0.1579
**A5** Left Rotate Right Rotate Mirror Roll/Pitch	x,y,z: (−0.6, 1.0, 5.0) m α: [−60, 60, 10]° β: [−60, 60, 10]° γ: 53.14°	x,y,z: (0.6, 1.0, 5.0) m α: [60, −60, −10]° β: [60, −60, −10]° γ: −53.14°	(#74) $\alpha_{l}$: −10°, $\beta_{l}$: 30°, $\alpha_{r}$: 10°, $\beta_{r}$: 30°,$\epsilon_{ctrl}$: 1.7785, $\epsilon_{corner}$: 0.1658 (#73) $\alpha_{l}$: −10°, $\beta_{l}$: 20°, $\alpha_{r}$: 10°, $\beta_{r}$: 20°,$\epsilon_{ctrl}$: 1.7820, $\epsilon_{corner}$: 0.1602 (#21) $\alpha_{l}$: −50°, $\beta_{l}$: 20°, $\alpha_{r}$: 50°, $\beta_{r}$: 20°, $\epsilon_{ctrl}$: 1.7852, $\epsilon_{corner}$: 0.1853 (#93) $\alpha_{l}$: 10°, $\beta_{l}$: −40°, $\alpha_{r}$: −10°, $\beta_{r}$: 40°,$\epsilon_{ctrl}$: 2.0763, $\epsilon_{corner}$: 0.1585

**Table 2 sensors-22-02067-t002:** Summary of the horizontal and vertical positioning experiments with dual checkerboards and their results. The best poses are defined by the top 2% lowest error of the projected control points. The subscripts *l* and *r* identify the left and right checkerboards, respectively. The # symbol in parenthesis represents the index of the experiment.

Experiment	Left Checkerboard Parameters	Right Checkerboard Parameters	Top 2% Poses, Its Control Points, and Corners Re-Projection Error ($\epsilon_{\mathrm{ctrl}},\epsilon_{\mathrm{corner}}$) [px]
**H1** Left Perpendicular Horizontally Together	α,β,γ: (0, 0, −53.14)° *x*: [−1.2, 0.4, 0.02] m *y*: 1.0 m *z*: 5.0 m	α,β,γ: (−40, −20, −53.14)° *x*: [−0.3, 1.3 0.02] m *y*: 1.0 m *z*: 5.0 m	(#74) $x_{l}$: 0.28 m, $x_{r}$: 1.18 m, $\epsilon_{ctrl}$: 1.0277, $\epsilon_{corner}$: 0.1751 (#78) $x_{l}$: 0.36 m, $x_{r}$: 1.26 m, $\epsilon_{ctrl}$: 1.0783, $\epsilon_{corner}$: 0.1739 (#15) $x_{l}$: −0.89 m, $x_{r}$: 0.0 m, $\epsilon_{ctrl}$: 1.3312, $\epsilon_{corner}$: 0.1227 (#5) $x_{l}$: −1.10 m, $x_{r}$: −0.02 m, $\epsilon_{ctrl}$: 1.3568, $\epsilon_{corner}$: 0.1268
**H2** Left/Right Mirrored Horizontally Together	α,β,γ: (50, −20, −53.14)° *x*: [−1.2, 1.3, 0.2] m *y*: 1.0 m *z*: 5.0 m	α,β,γ: (−50, 20, −53.14)° *x*: [−0.3, 0.4, 0.02] m *y*: 1.0 m *z*: 5.0 m	(#14) $x_{l}$: −0.92 m, $x_{r}$: −0.2 m, $\epsilon_{ctrl}$: 1.1834, $\epsilon_{corner}$: 0.144 (#75) $x_{l}$: 0.3 m, $x_{r}$: −1.2 m, $\epsilon_{ctrl}$: 2.0962, $\epsilon_{corner}$: 0.1826 (#17) $x_{l}$: −0.86 m, $x_{r}$: 0.04 m, $\epsilon_{ctrl}$: 2.1201, $\epsilon_{corner}$: 0.1440 (#18) $x_{l}$: −0.84 m, $x_{r}$: 0.06 m, $\epsilon_{ctrl}$: 2.1817, $\epsilon_{corner}$: 0.1488
**H3** Left Perpendicular Horizontally Separate	α,β,γ: (0, 0, −53.14)° *x*: [−0.4, −1.2, −0.01] m *y*: 1.0 m *z*: 5.0 m	α,β,γ: (−40, −20, −53.14)° *x*: [0.4, 1.2, 0.01] m *y*: 1.0 m *z*: 5.0 m	(#74) $x_{l}$: −1.14 m, $x_{r}$: 1.14 m, $\epsilon_{ctrl}$: 1.0383, $\epsilon_{corner}$: 0.1465 (#65) $x_{l}$: −1.05 m, $x_{r}$: 1.05 m, $\epsilon_{ctrl}$: 1.0566, $\epsilon_{corner}$: 0.1714 (#73) $x_{l}$: −1.13 m, $x_{r}$: 1.13 m, $\epsilon_{ctrl}$: 1.0842, $\epsilon_{corner}$: 0.1405 (#78) $x_{l}$: −1.18 m, $x_{r}$: 1.18 m, $\epsilon_{ctrl}$: 1.1105, $\epsilon_{corner}$: 0.1781
**H4** Left/Right Mirrored Horizontally Separate	α,β,γ: (50, −20, −53.14)° *x*: [−0.4, −1.2, −0.01] m *y*: 1.0 m *z*: 5.0 m	α,β,γ: (−50, 20, −53.14)° *x*: [0.4, 1.2, 0.01] m *y*: 1.0 m *z*: 5.0 m	(#20) $x_{l}$: −0.6 m, $x_{r}$: 0.6 m, $\epsilon_{ctrl}$: 1.7851, $\epsilon_{corner}$: 0.1853 (#43) $x_{l}$: −0.83 m, $x_{r}$: 0.83 m, $\epsilon_{ctrl}$: 1.9118, $\epsilon_{corner}$: 0.2020 (#54) $x_{l}$: −0.94 m, $x_{r}$: 0.94 m, $\epsilon_{ctrl}$: 1.9452, $\epsilon_{corner}$: 0.2076 (#31) $x_{l}$: −0.71 m, $x_{r}$: 0.71 m, $\epsilon_{ctrl}$: 2.0131, $\epsilon_{corner}$: 0.1901
**H5** Left/Right Mirrored Inverted Horizontally Separate	α,β,γ: (−50, 20, −53.14)° *x*: [−0.4, 1.2, 0.01] m *y*: 1.0 m *z*: 5.0 m	α,β,γ: (50, −20, −53.14)° *x*: [0.4,−1.2, 0.01] m *y*: 1.0 m *z*: 5.0 m	(#6) $x_{l}$: −0.92 m, $x_{r}$: −0.2 m, $\epsilon_{ctrl}$: 4.084, $\epsilon_{corner}$: 0.2147 (#14) $x_{l}$: 0.3 m, $x_{r}$: −1.2 m, $\epsilon_{ctrl}$: 4.7277, $\epsilon_{corner}$: 0.2521 (#13) $x_{l}$: −0.86 m, $x_{r}$: 0.04 m, $\epsilon_{ctrl}$: 4.7709, $\epsilon_{corner}$: 0.2238 (#23) $x_{l}$: −0.84 m, $x_{r}$: 0.06 m, $\epsilon_{ctrl}$: 5.43005, $\epsilon_{corner}$: 0.2391
**V1** Left/Right Identical Rotations Vertically Together	α,β,γ: (−40, −20, −53.14)° *x*: −1.13 m *y*: [0.35, 1.65, 0.02] m *z*: 5.0 m	α,β,γ: (−40, −20, −53.14)° *x*: 1.13 m *y*: [0.35, 1.65, 0.02] m *z*: 5.0 m	(#42) $y_{l}$: 1.19 m, $y_{r}$: 1.19 m, $\epsilon_{ctrl}$: 0.6763, $\epsilon_{corner}$: 0.1595 (#37) $y_{l}$: 1.09 m, $y_{r}$: 1.09 m, $\epsilon_{ctrl}$: 0.7285, $\epsilon_{corner}$: 0.1654 (#55) $y_{l}$: 1.45 m, $y_{r}$: 1.45 m, $\epsilon_{ctrl}$: 0.7542, $\epsilon_{corner}$: 0.1459 (#57) $y_{l}$: 1.49 m, $y_{r}$: 1.49 m, $\epsilon_{ctrl}$: 0.8238, $\epsilon_{corner}$: 0.1568
**V2** Left/Right Identical Rotations Vertically Opposite	α,β,γ: (−40, −20, −53.14)° *x*: −1.13 m *y*: [1.65, 0.35, −0.02] m *z*: 5.0 m	α,β,γ: (−40, −20, −53.14)° *x*: 1.13 m *y*: [0.35, 1.65, 0.02] m *z*: 5.0 m	(#34) $y_{l}$: 0.97m, $y_{r}$: 1.03 m, $\epsilon_{ctrl}$: 0.8916, $\epsilon_{corner}$: 0.1508 (#46) $y_{l}$: 0.73 m, $y_{r}$: 1.27 m, $\epsilon_{ctrl}$: 0.9385, $\epsilon_{corner}$: 0.1476 (#37) $y_{l}$: 0.91 m, $y_{r}$: 1.09 m, $\epsilon_{ctrl}$: 1.0684, $\epsilon_{corner}$: 0.1491 (#64) $y_{l}$: 0.37 m, $y_{r}$: 1.63 m, $\epsilon_{ctrl}$: 1.1009, $\epsilon_{corner}$: 0.1516

**Table 3 sensors-22-02067-t003:** Summary of the distance between camera and checkerboard experiments with dual checkerboards and their results. The best poses are defined by the top 2% lowest error of the projected control points. The subscript *l* and *r* identify the left and right checkerboards, respectively. The # symbol in parenthesis represents the index of the experiment.

Experiment	Left CheckerboardParameters	Right CheckerboardParameters	Top 2% Poses, Its Control Points, and Corners Re−Projection Error ($\epsilon_{\mathrm{ctrl}},\epsilon_{\mathrm{corner}}$) [px]
**D1** Left Fixed/>Right Approach	α,β,γ: (−40, −20, −53.14)° *x*: −1.13 m *y*: 1.0 m *z*: 5.0 m	α,β,γ: (−40, −20, −53.14)° *x*: 1.13 m *y*: 1.0 m *z*: [10.0, 4.6, −0.1] m	(#32) $z_{r}$: 6.8 m, $\epsilon_{ctrl}$: 1.2415, $\epsilon_{corner}$: 0.1671 (#43) $z_{r}$: 5.7 m, $\epsilon_{ctrl}$: 1.4161, $\epsilon_{corner}$: 0.1646 (#52) $z_{r}$: 4.8 m, $\epsilon_{ctrl}$: 1.4630, $\epsilon_{corner}$: 0.1638 (#54) $z_{r}$: 4.6 m, $\epsilon_{ctrl}$: 1.4680, $\epsilon_{corner}$: 0.1489
**D2** Left/Right Approach	α,β,γ: (−40, −20, −53.14)° *x*: −1.13 m *y*: 1.0 m *z*: [10.0, 4.6, −0.1]	α,β,γ: (−40, −20, −53.14)° *x*: 1.13 m *y*: 1.0 m *z*: [10.0, 4.6, −0.1]	(#49) $z_{l}$: 5.1 m, $z_{r}$: 5.1 m, $\epsilon_{ctrl}$: 1.3418, $\epsilon_{corner}$: 0.1898 (#30) $z_{l}$: 7.0 m, $z_{r}$: 7.0 m, $\epsilon_{ctrl}$: 1.5207, $\epsilon_{corner}$: 0.1872 (#50) $z_{l}$: 5.0 m, $z_{r}$: 5.0 m, $\epsilon_{ctrl}$: 1.5672, $\epsilon_{corner}$: 0.1541 (#46) $z_{l}$: 5.4 m, $z_{r}$: 5.4 m, $\epsilon_{ctrl}$: 1.5840, $\epsilon_{corner}$: 0.2177

**Table 4 sensors-22-02067-t004:** Summary of the simulation experiments with multiple checkerboards. The Experiment column contains the number of checkerboards and the details for each checkerboard per line. Position is expressed in meters, while rotations are expressed in degrees. The subscript *C* denotes the experiments of which the poses were manually selected following our guidelines. Numeric subscripts represent the top-performing poses.

Experiment (xyz),(rpy)	Image	Experiment (xyz),(rpy)	Image
**Three Checkerboards $\alpha_{1}$** (−0.05, 1.0, 4) m, (0, 0, −53.14)° (1.05, 1.4, 4.9) m, (10.07, 51.04, 40.53)° (−1.35, 0.65, 5.6) m, (34.04, −27.33, −35.99)°	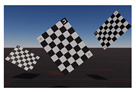	**Three Checkerboards $\alpha_{2}$** (−0.05, 1, 4) m, (0, 0, −53.14)° (1.05, 1.4, 4.9) m, (36.84, 42.16, −62.15)° (−1.35, 0.65, 5.6) m, (−33.57, 18.33, −6.1)°	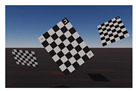
**Four Checkerboards $\beta_{1}$** (−0.45, 1.5, 5) m, (4.44, −14.28, −64.04)° (0.3, 0.7, 4.5) m, (−11.53, −38.45, −60.02)° (1.05, 1.4, 4.8) m, (10.07, 51.04, 40.53)° (−1.35, 0.65, 5.6) m, (34.04, −27.33, −35.99)°	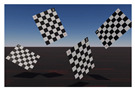	**Four Checkerboards $\beta_{2}$** (−0.45, 1.5, 5.0) m, (−11.53, −38.45, −60.02)° (0.3, 0.7, 4.5) m, (0, 0, −53.14)° (1.05, 1.4, 4.8) m, (10.07, 51.04, 40.53)° (−1.35, 0.65, 5.6) m, (34.04, −27.33, −35.99)°	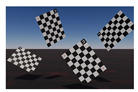
**Five Checkerboards $\gamma_{1}$** (0.05, 1, 4.1) m, (0, 0, −53.14)° (0.95, 1.6, 4.6) m, (−40.17, −20.68, −1.58)° (0.95, 0.65, 4.6) m, (10.07, 51.04, 40.53)° (−0.95, 1.5, 4.8) m, (−11.53, −38.45, −60.02)° (−0.75, 0.6, 4.3) m, (34.04, −27.33, −35.99)°	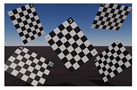	**Five Checkerboards $\gamma_{2}$** (0.05, 1, 4.3) m, (0, 0, −53.14)° (0.95, 1.6, 4.8) m, (−40.17, −20.68, −1.58)° (0.95, 0.65, 4.8) m, (10.07, 51.04, 40.53)° (−0.95, 1.5, 5) m, (−11.53, −38.45, −60.02)° (−0.75, 0.6, 4.5) m, (34.04, −27.33, −35.99)°	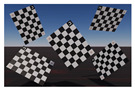
**Six Checkerboards $\delta_{1}$** (−0.05, 1.4, 4.4) m, (0, 0, −53.14)° (0.1, 0.6, 4.2) m, (−36.16, −43.53, −61.14)° (0.95, 1.6, 4.6) m, (20.17, −20.68, −1.58)° (0.95, 0.65, 4.6) m, (10.07, 51.04, 40.53)° (−0.95, 1.5, 4.7) m, (−11.53, −38.45, −60.02)° (−0.75, 0.6, 4.3) m, (34.04, −27.33, −35.99)°	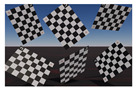	**Seven Checkerboards $\epsilon_{1}$** (−0.05, 1.7, 4.9) m, (−36.16, −43.53, 10.14)° (0.15, 1.0, 5.2) m, (0, 0, −53.14)° (0.1, 0.4, 4.7) m, (64.74, 8.31, −24.57)° (0.95, 1.6, 4.5) m, (20.17, −20.68, −1.58)° (0.95, 0.65, 4.5) m, (10.07, 51.04, 40.53)° (−0.95, 1.5, 4.7) m, (−11.53, −38.45, −60.02)° (−0.75, 0.6, 4.3) m, (34.04, −27.33, −35.99)°	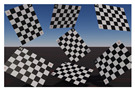
**Eight Checkerboards $\zeta_{1}$** (−1.15, 1.1, 5) m, (−11.53, −38.45, −60.02)° (−0.5, 1.7, 4.9) m, (−36.16, −43.53, 10.14)° (−0.6, 0.4, 5) m, (34.04, −27.33, −35.99)° (−0.35, 1.0, 5.4) m, (0, 0, −53.14)° (0.55, 0.95, 5) m, (40.93, 38.59, 36.58)° (0.5, 0.5, 4.4) m, (64.74, 8.31, 24.57)° (0.55, 1.59, 4.5) m, (20.17, −20.68, −1.58)° (1.35, 0.98, 5.05) m, (10.07, 51.04, 89.53)°	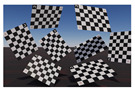	**Nine Checkerboards $\eta_{1}$** (−1.37, 0.9, 5.1) m, (−11.53, −58.45, −80.02)° (−0.8, 1.7, 4.9) m, (−36.16, −43.53, 10.14)° (−0.6, 0.4, 5.0) m, (34.04, −27.33, −35.99)° (−0.53, 1.08, 4.8) m, (−20.84, 26.11, −4.00)° (0.03, 1.65, 5.2) m, (0, 0, −53.14)° (0.4, 0.98, 4.65) m, (40.93, 38.59, 36.58)° (0.5, 0.5, 4.4) m, (64.74, 8.31, 24.57)° (0.75, 1.59, 4.5) m, (20.17, −20.68, −1.58)° (1.35, 0.98, 5.05) m, (10.07, 51.04, 89.53)°	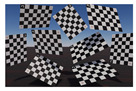
**Ten Checkerboards $\theta_{1}$** (−1.41, 1.05, 5.2) m, (−11.53, −58.45, −80.02)° (−0.8, 1.75, 5.0) m, (−36.16, −43.53, 10.14)° (−1.06, 0.32, 5.1) m, (39.04, −37.33, −15.99)° (−0.48, 1.0, 4.4) m, (−20.84, 26.11, −4.00)° (0.03, 1.65, 5.2) m, (0, 0, −53.14)° (0.4, 0.98, 4.65) m, (30.93, 28.59, 36.58)° (0.9, 0.4, 4.7) m, (64.74, 8.31, 24.57)° (0.89, 1.69, 4.9) m, (20.17, −20.68, −1.58)° (1.40, 0.95, 5.2) m, (10.07, 51.04, 89.53)° (−0.06, 0.3, 4.95) m, (64.74, 8.31, −24.57)°	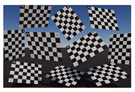		
**Six Checkerboards $\delta_{C}$** (0.05, 1.5, 5.1) m, (0, 0, −53.14)° (0.05, 0.30, 5.2) m, (10, −5, 0)° (−1.18, 1.3, 4.9) m, (9.23, −4.70, −1.58)° (1.18, 1.3, 5.1) m, (−4.72, 23.40, −15.88)° (1.29, 0.24, 5.2) m, (63.74, 18.31, −12.57)° (−1.38, 0.32, 5.3) m, (−22.92, −58.33, 5.44)°	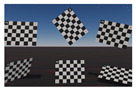	**Seven Checkerboards $\epsilon_{C}$** (0.05, 1.90, 5.5) m, (0, 5, 0)° (0.05, 1.05, 5.4) m, (0, 0, −53.14)° (0.05, 0.28, 5.0) m, (10, −5, 0)° (−1.18, 1.3, 4.9) m, (9.23, −4.70, −1.58)° (1.18, 1.3, 5.1) m, (−4.72, 23.40, −15.88)° (1.29, 0.24, 5.2) m, (63.74, 18.31, −12.57)° (−1.38, 0.32, 5.3) m, (−22.92, −58.33, 5.44)°	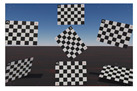

**Table 5 sensors-22-02067-t005:** Summary of the results for the simulation experiments with multiple checkerboards. The subscript *C* denotes the experiments of which the poses were manually selected following our guidelines. Numeric subscripts represent the top-performing poses.

Experiment	Distorted Corners RMSE [px]	Undistorted Corners RMSE [px]	Control Points RMSE [px]	Focal Length Error [px]	Center PointError [px]	DistortionError
Ten Checkerboards $\theta_{1}$	0.526	0.671	0.787	1.48	7.40	0.033
Nine Checkerboards $\eta_{1}$	0.515	0.830	0.888	8.344	9.91	0.076
Eight Checkerboards $\zeta_{1}$	0.519	0.708	1.01	1.58	9.47	0.025
Seven Checkerboards $\epsilon_{1}$	0.521	0.672	0.776	4.945	4.13	0.011
Six Checkerboards $\delta_{1}$	0.521	0.778	0.971	7.62	6.45	0.050
Five Checkerboards $\gamma_{2}$	0.494	0.947	1.22	3.89	11.5	0.065
Five Checkerboards $\gamma_{1}$	0.507	0.798	0.941	8.59	6.45	0.101
Four Checkerboards $\beta_{2}$	0.522	0.867	0.974	4.68	11.8	0.018
Four Checkerboards $\beta_{1}$	0.537	1.13	1.32	4.09	16.8	0.075
Three Checkerboards $\alpha_{2}$	0.524	0.651	0.693	32.5	21.4	0.0803
Three Checkerboards $\alpha_{1}$	0.508	0.762	0.808	9.79	1.72	0.0232
Six Checkerboards $\delta_{C}$	0.512	0.628	0.628	6.97	3.26	0.0092
Seven Checkerboards $\epsilon_{C}$	0.512	0.871	0.985	5.61	10.65	0.007

**Table 6 sensors-22-02067-t006:** List of the real-world cameras and its characteristics. HFOV stands for Horizontal Field of View.

Camera	Resolution	Sensor Size	Focal Length	HFOV
Lucid	2880 × 1860 (5.4 MP)	10.36 mm	8 mm	$56.7^{\circ }$
Lucid	2880 × 1860 (5.4 MP)	10.36 mm	25 mm	$19.6^{\circ }$
FLIR	1920 × 1260 (2.3 MP)	13.4 mm	8 mm	$69.7^{\circ }$

**Table 7 sensors-22-02067-t007:** Summary of the extrinsic parameters between the cameras and the 3D LiDAR for the outdoors dataset: x,y,z are in meters; roll, pitch and yaw are in radians. We truncated the floating-point values to 3 digits for formatting purposes. The **W** next to the camera name stands for wide, while the **T** stands for Telephoto, representing the measurements obtained with the 8 and 25 mm lenses, respectively.

Experiment	x	y	z	roll	pitch	yaw
**Lucid W$\delta_{1}$**	0.031	−0.14	0.03	−1.515	0.005	−1.628
**Lucid W$\delta_{C}$**	0.031	−0.14	0.03	−1.474	0.01	−1.625
**Lucid W$\epsilon_{1}$**	0.031	−0.14	0.05	−1.513	−0.01	−1.619
**Lucid W$\epsilon_{C}$**	0.031	−0.12	0.05	−1.51	−0.01	−1.623
**FLIR W$\delta_{1}$**	0.031	−0.14	0.03	−1.5	0.005	−1.564
**FLIR W$\delta_{C}$**	0.031	−0.14	0.03	−1.48	0.005	−1.564
**FLIR W$\epsilon_{1}$**	0.031	−0.14	0.03	−1.5	0.005	−1.564
**FLIR W$\epsilon_{C}$**	0.031	−0.12	0.03	−1.504	−0.02	−1.563
**Lucid T$\delta_{1}$**	0.05	−0.14	0.031	−1.588	−0.05	−1.509
**Lucid T$\delta_{C}$**	0.05	−0.14	0.031	−1.559	−0.049	−1.521
**Lucid T$\epsilon_{1}$**	0.05	−0.14	0.031	−1.553	−0.045	−1.517
**Lucid T$\epsilon_{C}$**	0.05	−0.15	0.031	−1.548	−0.05	−1.515

**Table 8 sensors-22-02067-t008:** Summary of the real-world experiments with multiple checkerboards on two different cameras with wide angle lenses.

Experiment	Image
**Lucid Six Checkerboards $\delta_{1}$**	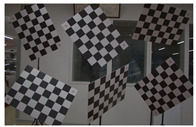
**Lucid Six Checkerboards $\delta_{C}$**	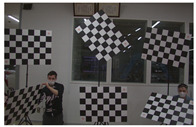
**Lucid Seven Checkerboards $\epsilon_{1}$**	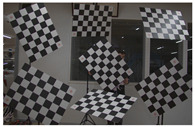
**Lucid Seven Checkerboards $\epsilon_{C}$**	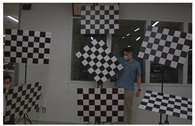
**FLIR Six Checkerboards $\delta_{1}$**	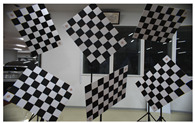
**FLIR Six Checkerboards $\delta_{C}$**	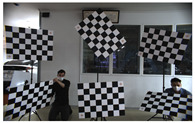
**FLIR Seven Checkerboards $\epsilon_{1}$**	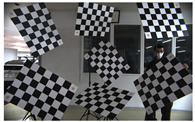
**FLIR Seven Checkerboards $\epsilon_{C}$**	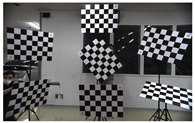

**Table 9 sensors-22-02067-t009:** Summary of the real-world experiments with multiple checkerboards with a telephoto lens.

Experiment	Image
**Lucid Six Checkerboards $\delta_{1}$**	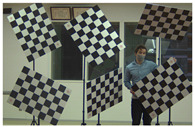
**Lucid Six Checkerboards $\delta_{C}$**	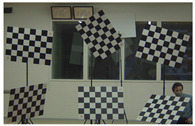
**Lucid Seven Checkerboards $\epsilon_{1}$**	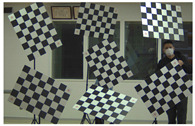
**Lucid Seven Checkerboards $\epsilon_{C}$**	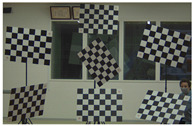

## Data Availability

Not applicable.

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
