# Peer review of "Single-Shot Intrinsic Calibration for Autonomous Driving Applications"

_sensors, 2022, doi:10.3390/s22052067_

Round 1

Reviewer 1 Report

can the algorithm be applied in real-time?

any quantitative analysis on results?

Author Response

Q1:can the algorithm be applied in real-time?

Author response: Thank you for reading our work, and we appreciate your kind comment. Even if our approach is not meant to be executed in real-time, since we aim for higher accuracy on both the corner detection and the estimation of the calibration parameters, the results can be obtained in a few seconds after taking a single image. 

We have added a paragraph indicating this fact to clarify the execution times of our method.

Q2: any quantitative analysis on results?

Author response: You can find a thorough analysis in Tables 1,2,3 and summarized results in Table 5 and Figure 6. We understand it is a long paper and might be difficult to follow. Thanks to your valuable comment, we have added pointers in Section 3.6.3 to find the key sections of the paper quickly.

Reviewer 2 Report

This is an well written and highly detailed study focused on developing a proper methodology for reliabile estimation of camera intrinsic parameters for 3D applications. The authors have proved the novelty and repeatability of their proposed methodology by detailed evalution on both real-world and synthetic enviroment set. The evaluated guidelines will prove to be practically helpful to the community. Additionally, the figures are also quite clear and easy to read. I don't have any significant correction comments. Very good work !

Author Response

Q1: This is an well written and highly detailed study focused on developing a proper methodology for reliabile estimation of camera intrinsic parameters for 3D applications. The authors have proved the novelty and repeatability of their proposed methodology by detailed evalution on both real-world and synthetic enviroment set. The evaluated guidelines will prove to be practically helpful to the community. Additionally, the figures are also quite clear and easy to read. I don't have any significant correction comments. Very good work !

Author response: We appreciate your kind comments. Thank you for reading the paper. We understand it is on the long side. We put a lot of effort while writing it and, we feel delighted thanks to your words. We aim to produce more practical and helpful research for the community. We additionally ran an additional check and removed some misspellings. Thank you for catching them.

Reviewer 3 Report

The paper address the camera calibration method to determine clear and repeatable guidelines for single-shot camera intrinsic calibration 
using multiple checkerboards. With the help of a simulator, the position and rotation intervals that allow optimal corner detectors can be founded.

Q1. Although camera calibration for autonomous vehicle development is the subject of this paper, it is reasonable to delete the autonomous driving applications from the title of the paper, as it does not actually include autonomous driving application experiments.

Q2. [Line 28] Among the four references cited on sensor fusion, [1] Complex YOLO paper is not related to sensor fusion, so it is reasonable to exclude it.

Q3. In order to help intuitive understanding of the 3D coordinate system, it is suggested to display the 3D coordinate system in Figure 1.

Q4. [Line 212 ~ 216] Add definitions for C, P, K, Dt Vector described in this sentence.

Q5. In Figure 4, the long-range Control Point is set to 50m. In a real telephoto lens, 50m is thought to be closer to near, not far.
For this reason, telephoto results seem to be relatively bad. I would like to suggest experimenting with different Control Point distances.

Q6. Please correct the error in the subscript indicating the number of checkerboards in Table 8 and Table 9.

Author Response

Q1. Although camera calibration for autonomous vehicle development is the subject of this paper, it is reasonable to delete the autonomous driving applications from the title of the paper, as it does not actually include autonomous driving application experiments. 

Author response:  We apologize for the lack of context about autonomous driving. We added a paragraph in the introduction about why we decided to include autonomous driving applications in the title. In our experience, robots or vehicles featuring large arrays of cameras present a time-consuming process when calibrating multiple cameras. Especially when using the obtained parameters to project 3D sensors such as LiDARs or Radars. For that reason, we envisioned a simplified and automated method to calibrate the cameras with reduced effort and reduce the chance of human error while manipulating the calibration targets. 

Q2. [Line 28] Among the four references cited on sensor fusion, [1] Complex YOLO paper is not related to sensor fusion, so it is reasonable to exclude it. 

Author response:  

Thank you for carefully reading the paper. We completely agree with your comment. We replaced the Complex-YOLO paper, a point cloud only detection, with PointPainting, another well-known multi-modal object detection method that requires accurate calibration.

Replaced the indicated reference.

Q3. In order to help intuitive understanding of the 3D coordinate system, it is suggested to display the 3D coordinate system in Figure 1.

Author response:  

We appreciate your suggestion and agree with the change.

We added the checkerboard coordinate system to the figure.

Q4. [Line 212 ~ 216] Add definitions for C, P, K, Dt Vector described in this sentence.

Author response:  

We agree with your suggestion, this will help to clarify the meaning of these matrices.

We added the definition for the mentioned matrices.

Q5. In Figure 4, the long-range Control Point is set to 50m. In a real telephoto lens, 50m is thought to be closer to near, not far.

For this reason, telephoto results seem to be relatively bad. I would like to suggest experimenting with different Control Point distances.

Author response:  We agree with your comment and added explanatory sentences about the selection of control points.

We followed the approach to simulate a wide-angle lens, evaluate the poses synthetically, select the best results, and test them again on a real camera. For that reason, we initially chose 50m as long-range. Since we found that the approach worked well, we were curious about how the same configuration on a telephoto camera would perform. However, as you correctly pointed out, it is uncertain if the same quantitative evaluation would be valid on a telephoto lens due to the changes in FOV and distance. We found that appropriate control point design is essential thanks to these results. As mentioned in the conclusions, we plan to include simulation experiments with other camera types to find suitable configurations.

We have added a paragraph in the conclusions stating this fact.

Q6. Please correct the error in the subscript indicating the number of checkerboards in Table 8 and Table 9.

Author response:  

We apologize for the mistake. We modified the subscripts to match the experiments’ subscripts.

Round 2

Reviewer 3 Report

Accept in present form.